# Utility of machine learning in developing a predictive model for early-age-onset colorectal neoplasia using electronic health records

Hisham Hussan[1,2]*, Jing Zhao[3], Abraham K. Badu-Tawiah[1,4,5], Peter Stanich[1], Fred Tabung[2,6], Darrell Gray[1,2], Qin Ma[3], Matthew Kalady[2,7], Steven K. Clinton[2,6]

1 Division of Gastroenterology, Hepatology, and Nutrition, Department of Internal Medicine, The Ohio State University, Columbus, Ohio, United States of America, 2 Comprehensive Cancer Center, The Ohio State University, Columbus, Ohio, United States of America, 3 Department of Biomedical Informatics, College of Medicine, The Ohio State University, Columbus, Ohio, United States of America, 4 Department of Chemistry and Biochemistry, The Ohio State University, Columbus, Ohio, United States of America, 5 Department of Microbial Infection and Immunity, The Ohio State University, Columbus, Ohio, United States of America, 6 Division of Medical Oncology, Department of Internal Medicine, College of Medicine, The Ohio State University, Columbus, Ohio, United States of America, 7 Division of Colon and Rectal Surgery, Department of Surgery, The Ohio State University, Columbus, Ohio, United States of America

* Hisham.hussan@osumc.edu

## Abstract

### Background and aims

The incidence of colorectal cancer (CRC) is increasing in adults younger than 50, and early screening remains challenging due to cost and under-utilization. To identify individuals aged 35–50 years who may benefit from early screening, we developed a prediction model using machine learning and electronic health record (EHR)-derived factors.

### Methods

We enrolled 3,116 adults aged 35–50 at average-risk for CRC and underwent colonoscopy between 2017–2020 at a single center. Prediction outcomes were (1) CRC and (2) CRC or high-risk polyps. We derived our predictors from EHRs (e.g., demographics, obesity, laboratory values, medications, and zip code-derived factors). We constructed four machine learning-based models using a training set (random sample of 70% of participants): regularized discriminant analysis, random forest, neural network, and gradient boosting decision tree. In the testing set (remaining 30% of participants), we measured predictive performance by comparing C-statistics to a reference model (logistic regression).

### Results

The study sample was 55.1% female, 32.8% non-white, and included 16 (0.05%) CRC cases and 478 (15.3%) cases of CRC or high-risk polyps. All machine learning models predicted CRC with higher discriminative ability compared to the reference model [e.g., C-statistics (95%CI); neural network: 0.75 (0.48–1.00) vs. reference: 0.43 (0.18–0.67); P = 0.07]

**Data Availability Statement:** All relevant data are within the manuscript and its Supporting Information files.

**Funding:** This work was supported by an award (UL1TR002733) from the National Center for Advancing Translational Sciences. The content is solely the responsibility of the authors and does not necessarily represent the official views of the National Center for Advancing Translational Sciences or the National Institutes of Health.

**Competing interests:** NO authors have competing interests.

**Abbreviations:** AUC, area under the curve; AUROC, area under the receiver operating characteristic curve; ASA, American Society of Anesthesiology; BMI, body mass index; CRC, colorectal cancer; EHR, electronic health record; FIT, fecal immunochemical test; HDL, high-density lipoprotein; LDL, low densitiy lipoprotein; IRB, institutional review board; RUCA, rural-urban commuting area; WHO, World Health Organizatio.

Furthermore, all machine learning approaches, except for gradient boosting, predicted CRC or high-risk polyps significantly better than the reference model [e.g., C-statistics (95%CI); regularized discriminant analysis: 0.64 (0.59–0.69) vs. reference: 0.55 (0.50–0.59); P<0.0015]. The most important predictive variables in the regularized discriminant analysis model for CRC or high-risk polyps were income per zip code, the colonoscopy indication, and body mass index quartiles.

## Discussion

Machine learning can predict CRC risk in adults aged 35–50 using EHR with improved discrimination. Further development of our model is needed, followed by validation in a primary-care setting, before clinical application.

## Introduction

Colorectal cancer (CRC) is the most gastrointestinal cancer, affecting over 150,000 adults in the U.S. each year. Despite a declining CRC incidence and mortality in older adults due to effective screening, CRC incidence and mortality is rising in adults ≤50 years of age [1–3]. The duration of preclinical CRC is estimated to be between 4 and 6 years [4]. Thus, adults may harbor asymptomatic CRC for years before undergoing CRC screening [5]. Although multiple professional societies now recommend initiating CRC screening at age 45 as opposed to 50, simulations raise concerns about cost, risks, and efficacy even when fecal immunochemical testing (FIT) is used [6]. Thus, there is an urgent need to establish novel and targeted CRC screening strategies for young adults that are cost-effective and easy to implement. Such efforts are challenged by the perceived lower risk among young adults and medical providers, even when gastrointestinal symptoms are present [7–9].

One potential strategy for the early detection of CRC and premalignant polyps, is to apply evidence-based risk stratification tools to identify individuals at greater risk who can benefit from screening. Such efforts may reduce diagnostic delays for young adults, particularly when leveraging the power of electronic medical records to alert caregivers. Novel risk assessment tools are being developed and validated for other malignancies and applied to clinical practice so as to improve care with acceptable costs [10]. However, thus far, the available CRC risk assessment tools focus on asymptomatic adults over the age of 50 and do not capture adults aged 35–44 who account for 50% of early-onset CRC cases [11, 12]. Existing CRC prediction tools also lack discriminatory power or are cumbersome to use, which has reduced their utilization and dissemination [13]. Therefore, developing a sensitive and specific, and yet easy to implement, CRC risk assessment tool for adults aged 35–50 is necessary to classify young adults into meaningful risk groups so as to identify those at high risk, while reducing interventions such as colonoscopy, in those at low risk.

In that regard, machine learning is an aspect of artificial intelligence that uses software algorithms to improve the analysis by learning and identifying patterns in large datasets [14]. Therefore, Incorporation of machine learning offers potential for the development of an effective CRC risk assessment tool for young adults. For instance, machine learning methods that integrate clinical risk factors have been applied to breast cancer risk prediction and improve predictive accuracy from 60% to 90% [15]. In addition, deep learning with an artificial neural network based on personal health data has been shown to robustly stratify CRC risk in the

large national database [16]. Thus, we hypothesize that machine learning can integrate readily available and complex factors from electronic health records (EHRs) to create a prediction model for CRC that applies to adults aged 35–50. To test our hypothesis, we derived and internally validated a prediction model for CRC or high-risk polyps in adults aged 35–50 years who underwent colonoscopy due to symptoms or screening age indications.

## Methods

### Participants

We conducted a retrospective predictive study at the Ohio State University after receiving approval from the Ohio State University's (OSU) Institutional Review Board (IRB protocol number 2020H0190). The Ohio State University IRB approved the waiver of informed consent since this is a retrospective chart review that involves no interaction with study participants; and the study accessed information which would normally be accessed during clinical care for these patients. To develop the model, we used data from average-risk adults aged 35–50 who underwent their first colonoscopy between November 2017 and February 2020. Our cohort and their colonoscopy data were obtained from the GI Quality Improvement Consortium (GIQuic) database at the Ohio State University. GIQuic is a collaborative, nonprofit, scientific organization between the American College of Gastroenterology and the American Society for Gastrointestinal Endoscopy [17]. The OSU GIQuIC database collects all the colonoscopies performed at OSU and includes colonoscopy quality measures such as adequacy of bowel preparation, indication, cecal intubation rate, and adenoma detection rate [18].

### Inclusion and exclusion criteria

Our study plot is included in Fig 1. We included adults aged 35–50 years due to the increased incidence of early-onset CRC in adults aged 35–49 years compared to younger adults [1]. Moreover, a substantial portion of asymptomatic early-onset CRC patients are not diagnosed until the initiation of screening at age 50 [5]. Some health plans previously approved CRC screening colonoscopy in adults aged 45 and older according with the American Cancer Society (ACS) 2018 guidelines [19]. However all patients <45 are generally referred for a diagnostic colonoscopy (e.g., diarrhea, constipation, abdominal pain, irritable bowel syndrome, bleeding, etc). These gastrointestinal symptoms are found in a significant proportion of Americans, most of whom do not undergo diagnostic colonoscopy or have no organic causes on a colonoscopy [20–23]. Therefore, we included adults who underwent either a diagnostic or screening colonoscopy, which is the standard of care for the diagnosis of polyps or CRC. We further investigated if including diagnostic colonoscopy may lead to possible bias by comparing our predictors between adults aged 46–49 undergoing screening colonoscopy to vs. diagnostic colonoscopy (S1 Table). We selected 46–49 because the numbers of diagnostic and screening colonoscopies were similar in this age range (407 vs. 296, respectively). Only diagnostic symptoms, tobacco use, and triglyceride levels differed significantly, suggesting symptomatic and asymptomatic adults are similar for most of the predictors included in this study. All included adults had a complete colonoscopy and an adequate bowel prep for detection of polyps >5 mm (Boston Bowel Prep Scale ≥2 in every colon segment) [24]. As early-onset CRC primarily occurs in adults with no strong familial predisposition or pre-existing colitis [25], we included only average-risk adults in our model. Of the 5,588 participants considered for the study, we excluded patients with: (1) inflammatory bowel disease or colitis on subsequent biopsies; (2) personal history of polyps or CRC, elevated cancer makers (e.g., CEA or CA199) or metastatic cancer requiring colonoscopy; (3) family history of CRC in one first degree or two second degree relatives, or (4) hereditary CRC syndromes including polyposis

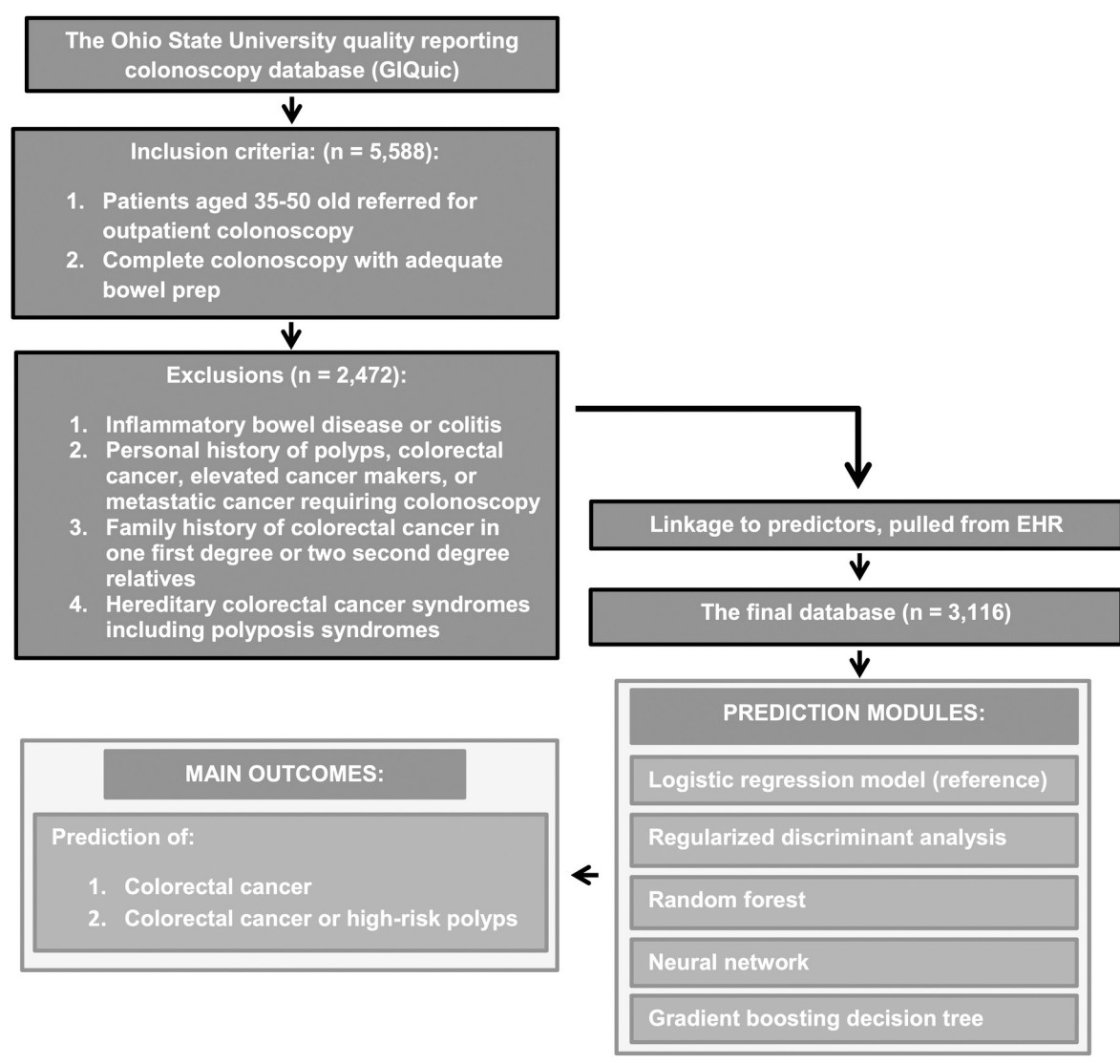

**Fig 1. Study plot detailing study flow as well as inclusion and exclusion criteria.**

syndromes. After these exclusions, we retained 3,116 participants. This study was conducted and reported in accordance with the guidelines for transparent reporting of a multivariable prediction model for individual prognosis or diagnosis (TRIPOD) [26].

## Outcomes

Our two outcomes were (1) CRC and (2) CRC or high-risk polyps confirmed by pathology. Our outcomes were recorded by a personnel who was blinded to the study patient clinical characteristics and laboratory measurements. High-risk polyps were defined as adenomatous or serrated polyps ≥10 mm in size with high grade dysplasia or villous component or ≥3 adenomas or serrated polyps of any size as done before [27, 28]. We did not include hyperplastic polyps in our outcomes due to benign nature. Because the rate of early-onset CRC is low (9.5–14 per 100,000) [29], we included high-risk polyps as an outcome to increase the pre-test probability of a positive test. We chose to include only high-risk polyps as 91% of proven high-risk

polyps grow by 20% per year, whereas 63% of non-high-risk polyps remain stable or regress [30].

## Predictors

The predictors for machine learning models were chosen from routinely available data in the EHRs using *a priori* knowledge. Our predictors were pulled from EHRs by a data analyst who was blinded to the study and our outcomes. The predictors included in this study are summarized in Table 1. Predictors included patient age at time of colonoscopy, reported sex and race; American Society of Anesthesiology (ASA) comorbidity category; the symptom indicating the colonoscopy; recorded height and weight; calculated body mass index (BMI) classified by mean, quartile, and overweight/obesity category per WHO criteria; social history (use of alcohol, tobacco, or recreational drugs); medication (aspirin and statins); and laboratory studies (hemoglobin and cholesterol panels). We used the ratio of triglyceride to high-density lipoprotein (HDL) as a surrogate of insulin resistance and stratified the ratio by mean, quartile, and $<3$ or $\geq 3$ as previously reported [31, 32]. Zip code-derived social determinants of health were retrieved as well. Specifically, we used the 2018 IRS public data to link zip codes to mean adjusted gross income, adjusted gross income percentage within income brackets, and single return percentile [33] for inclusion in our model. Rural-urban commuting area (RUCA) codes, which are a detailed and flexible scheme for delineating sub-county components of rural and urban areas, were also linked to zip codes and included as predictors in our model [34].

## Data preparation

Dummy variables were created by converting categorical variables to corresponding as many numerical variables as there are categories. Then, training and testing datasets were generated by randomly splitting the data into 70% and 30%, respectively, using the *createDataPartition* function in the R package *caret*. Missing values were imputed by creating a bag imputation model and using the imputation model to predict the values of missing data points. Imputation via bagging fits a bagged tree model for each predictor as a function of all other predictors. Finally, we centered, scaled, and transformed the predictor values using the R function *preProcess* in the *caret* package to generate comparable continuous predictors with dynamic ranges.

## Predictive modeling

To predict the probability of each outcome, we first fit a logistic regression model as the reference model including all of the aforementioned predictors. Then four machine learning models were constructed: (1) regularized discriminant analysis, (2) random forest, (3) neural network, and (4) gradient boosting decision tree. Regularized discriminant analysis is a generalization of linear discriminant analysis and quadratic discriminant analysis that increases the power of discriminant analysis to penalize large coefficients from small sample sizes. For our regularized discriminant analysis model, we performed a random search for two parameters (gamma and lambda) using the R package *klaR* [35]. Random forests are a bagging approach derived from many decision trees and are created with bootstrap samples of training data and random feature selection. For our random forest model, we used random search in the *randomForest* package to generate 20 random values of *mtry* and selected the value with the highest accuracy [36]. Neural networks are computational learning systems that use a network of functions to understand and translate a data input of one form into a desired output. For our neural network model, we performed a random search for two hyper-parameters (size and decay) using the R package *nnet* [37]. Gradient boosting decision trees are a boosting approach

**Table 1. Included predictors and baseline demographics.**

| Predictors | Percentages and means | Missing data |
|---|---|---|
| Total number of included patients | 3116 | |
| Mean age (standard deviation or S.D.) | 46.5 (4.73) | 0.0% |
| Female Gender | 55.1% | 0.0% |
| Race | | 1.8% |
| Non-Hispanic White | 67.2% | |
| African American | 18.7% | |
| Hispanic | 2.6% | |
| Asian | 1.5% | |
| Other | 8.1% | |
| Rural-urban commuting area code (RUCA2) | | 0.0% |
| Mean (S.D.) | 1.33 (1.17) | |
| Metropolitan [RUCA 1–3] | 93.3% | |
| Micropolitan [RUCA 4–6] | 5.0% | |
| Small town [RUCA 7–9] | 1.4% | |
| Rural [RUCA 10] | 0.3% | |
| Percentage of returns within income brackets per zip code [mean (S.D.)] | | 0.4% |
| $1 to under $25,000 | 30.60% (9.48) | |
| $25,000 to under $50,000 | 24.59% (6.98) | |
| $50,000 to under $75,000 | 14.64% (2.63) | |
| $75,000 to under $100,000 | 9.50% (2.83) | |
| $100,000 to under $200,000 | 14.70% (7.69) | |
| $200,000 or more | 5.97% (5.88) | |
| Percentage of single tax returns per Zip code [mean (S.D.)] | 49.04% (8.67) | 0.4% |
| Adjusted gross income per zip code [mean (S.D.)] | $1,399,101.15 (862,516.76) | 0.4% |
| American Society of Anesthesiology (ASA) Physical Status Classification System | | 0.0% |
| ASA I (healthy patient) | 25.6% | |
| ASA II (mild systemic disease) | 67.4% | |
| ASA III (severe systemic disease) | 6.9% | |
| ASA IV (life threatening systemic disease) | 0.1% | |
| Colorectal cancer screening indication | 53.0% | 0.0% |
| All diagnostic colonoscopy indications | 47.0% | 0.0% |
| Functional gastrointestinal symptoms: | 32.8% | |
| •Abdominal pain | 11.3% | |
| •Constipation | 5.9% | |
| •Diarrhea | 3.3% | |
| •Rectal pain | 0.6% | |
| •Pelvic pain | 0.3% | |
| •Obstipation | 0.1% | |
| •Irritable bowel syndrome | 0.3% | |
| Weight loss | 1.0% | |
| Gastrointestinal bleeding | 20.1% | |
| Anemia | 3.8% | |
| Change in bowel habits | 2.8% | |
| Change in stool caliber | 0.7% | |
| Personal history of cancer other than CRC | 0.4% | |
| Colorectal neoplasm in distant relative | 3.0% | |

*(Continued)*

**Table 1.** (Continued)

| Predictors | Percentages and means | Missing data |
|---|---|---|
| Family history of cancer other than CRC | 0.0% | |
| Prior diverticulitis prior diverticulitis | 2.1% | |
| Height in feet [mean (S.D.)] | 5.59 (0.34) | 0.7% |
| Weight in pounds [mean (S.D.)] | 194.58 (53.11) | 6.0% |
| BMI (kg/m$^2$) | | 6.0% |
| Mean (S.D.) | 30.24 (7.42) | |
| $\geq$ 25 Kg/m$^2$ | 70.7% | |
| $\geq$ 30 Kg/m$^2$ | 40.5% | |
| $\geq$ 35 Kg/m$^2$ | 19.4% | |
| $\geq$ 40 Kg/m$^2$ | 9.4% | |
| Median [Inter quartile Range (IQR)] | 28.8 (25–33.9) | |
| Alcohol use | | 0.9% |
| Never | 1.1% | |
| No | 33.8% | |
| Not currently | 3.0% | |
| Yes | 61.2% | |
| Tobacco use | | 0.4% |
| Never | 61.9% | |
| Passive | 0.2% | |
| Quit | 23.1% | |
| Yes | 14.3% | |
| Intravenous drug user | | 1.4% |
| No | 98.4% | |
| Yes | 0.2% | |
| Illicit drug user | | 1.4% |
| Never | 5.5% | |
| No | 84.3% | |
| Not currently | 2.0% | |
| Yes | 6.8% | |
| Total cholesterol (mg/dL) | | 29.2% |
| Mean (S.D.) | 185.57 (41.11) | |
| $\geq$ 200 mg/dL | 23.9% | |
| < 200 mg/dL | 46.8% | |
| $\geq$ 170 mg/dL | 45.3% | |
| < 170 mg/dL | 25.5% | |
| Median (IQR) | 183 (159–210) | |
| High Density Lipoprotein (HDL, mg/dL) | | 29.9% |
| Mean (S.D.) | 51.90 (15.82) | |
| $\geq$35 mg/dL | 63.5% | |
| <35 mg/dL | 6.6% | |
| $\geq$40 mg/dL | 55.4% | |
| <40 mg/dL | 14.7% | |
| Median (IQR) | 49 (41–60) | |
| Low Density Lipoprotein (LDL, mg/dL) | | 30.3% |
| Mean (S.D.) | 107.01 (34.45) | |
| $\geq$100 mg/dL | 40.3% | |
| <100 mg/dL | 29.4% | |

(*Continued*)

**Table 1.** (Continued)

| Predictors | Percentages and means | Missing data |
|---|---|---|
| ≥150 mg/dL | 7.2% | |
| <150 mg/dL | 62.5% | |
| Median (IQR) | 106 (84–129) | |
| Triglyceride (TG, mg/dL) | | 29.4% |
| Mean (S.D.) | 143.74 (176.88) | |
| ≥150 mg/dL | 21.9% | |
| <150 mg/dL | 48.7% | |
| *Median (IQR)* | 110 (76–167) | |
| Triglyceride: High Density Lipoprotein (TG: HDL) ratio | | 29.9% |
| Mean (S.D.) | 3.31 (5.92) | |
| High (ratio ≥3) | 25.0% | |
| Low (ratio <3) | 45.1% | |
| *Median (IQR)* | 2.24 (1.35–3.77) | |
| Hemoglobin (mg/dL) | | 40.3% |
| Mean (S.D.) | 13.75 (1.71) | |
| Females with anemia (<12 mg/dL) | 6.3% | |
| Males with anemia (<13.5 mg/dL) | 3.2% | |
| *Median (IQR)* | 13.9 (12.8–14.9) | |
| Reported non-steroidal anti-inflammatory drugs use | 12.5% | 0.0% |
| Statin medications use | 14.3% | 0.0% |

that builds an additive model of decision trees estimated by gradient descent. For our gradient boosting decision tree model, we applied a random search to tune parameters, number of iterations, and interaction depth while holding shrinkage constant in the R package *gbm* [38]. Finally, to account for potential overfitting of the machine learning models, we employed repeated five-fold cross-validation in the R package *caret* [34].

We assessed the predictive performance of each model by computing C-statistics [area under the receiver operating characteristic curve (AUC/AUROC)] and prediction metrics, including sensitivity, specificity, positive predictive value, and negative predictive value, in the testing dataset using functions provided in the R package *pROC* [39]. To account for the class imbalance caused by a low proportion of outcomes, we applied the SMOTE resampling method to generate artificial samples [40] and selected cutoffs based on the ROC curve. To evaluate the contribution of each predictor to the machine learning models, we calculated variable importance in the best performing models. Finally, we used the DeLong test to compare ROC curves to the reference model using the function *roc.test* in the R package *pROC* [39]. A p-value <0.05 was considered statistically significant. All analyses were performed with R version 4.0.2 (The R Foundation for Statistical Computing).

## Results

Altogether, 3,116 adults aged 35–50 were included in our study. The characteristics of the participants are described in Table 1. The cohort was 55.1% female, 32.8% non-white, and 93.3% belonged to a metropolitan area per RUCA classifications. Approximately 54% of the cohort belonged to zipcode that earned less than $50,000 a year, and more than two-thirds (72%) of the cohort were overweight or obese. A screening colonoscopy was performed in 53% of participants, and functional gastrointestinal symptoms were the main indication for a diagnostic colonoscopy (32.8%).

## Prediction of colorectal cancer

Overall, 16 (0.05%) patients had CRC on colonoscopy. The C-statistics for CRC are presented as ROC curves in Fig 2 and comparisons of AUC characteristics in Fig 3. The reference model had the lowest discriminative ability when compared to the machine learning models. For example, the neural network and gradient boosting decision tree models had higher AUC values compared to the reference model but did not reach significance (neural network: 0.75; 95%CI, 0.48–1.00; $P = 0.07$; stochastic gradient boosting: 0.76; 95%CI, 0.46–1.00; $P = 0.14$). The performance metrics for all models are detailed in Table 2. All machine learning models had similar sensitivity to the reference model except regularized discriminant analysis, which had lower sensitivity. The specificity and accuracy were much higher for machine learning models compared to the reference model (e.g., specificity of 96% and accuracy of 96% for regularized discriminant analysis vs. 10% and 11%, respectively, for the reference model). The balanced accuracy, a better metric for a rare event like CRC, was higher in the machine learning models compared to the reference model (e.g., 73% for the neural network vs. 42% for the reference). Due to the low incidence of CRC, the positive predictive value was low in all models (maximum of 5% for regularized discriminant analysis), and the negative predictive value was 99% in all models.

## Prediction of colorectal cancer or high-risk polyps

There were 478 (15.34%) participants with CRC or high-risk polyps in our cohort. The C-statistics are described in Figs 2 and 3. All machine learning approaches had a significantly higher ability to predict CRC or high-risk polyps, except for gradient boosting decision tree. For example, the regularized discriminant analysis model had an AUC of 0.64 (95%CI, 0.59–0.69), whereas the reference model had an AUC of 0.55 (95%CI, 0.5–0.6; P<0.0015). Compared with the reference model, all machine learning models had comparable sensitivity and slightly higher specificity, accuracy, and balanced accuracy (Table 2). The positive predictive value was higher in the machine learning models compared to the reference model (e.g., 0.27 for the regularized discriminant analysis model vs. 0.17 for the reference model). The negative predictive value was comparable in all models with a maximum of 0.89 in the regularized discriminant analysis and neural network models.

## Variable importance

The importance of variables in predicting the risk of CRC and CRC or high-risk polyps is demonstrated in Fig 4. These variables were calculated in the best performing models based on our AUC comparisons to reference illustrated in Fig 3: The neural network for CRC and regularized discriminant analysis for CRC or high-risk polyps. The leading predictors of CRC in the neural network model were ASA comorbidity category, HDL quartile, gastrointestinal bleeding as an indication for diagnostic colonoscopy, mean percent of single returns per zip code, BMI quartiles, and mean triglyceride:HDL ratio. The most important predictive variables for CRC or high-risk polyps in the regularized discriminant analysis model were income returns within income brackets per zip code, the indication for colonoscopy (screening vs. diagnostic), BMI quartiles, triglyceride:HDL ratio (<3 vs ≥3), alcohol use, ASA comorbidity category, statin use, HDL category, and gastrointestinal bleeding as an indication for diagnostic colonoscopy.

## Discussion

Cost-effective tools are needed to improve CRC screening in young adults and to reduce the morbidity and mortality associated with delayed diagnosis. We assessed the utility of machine

**CRC: ROC curves**

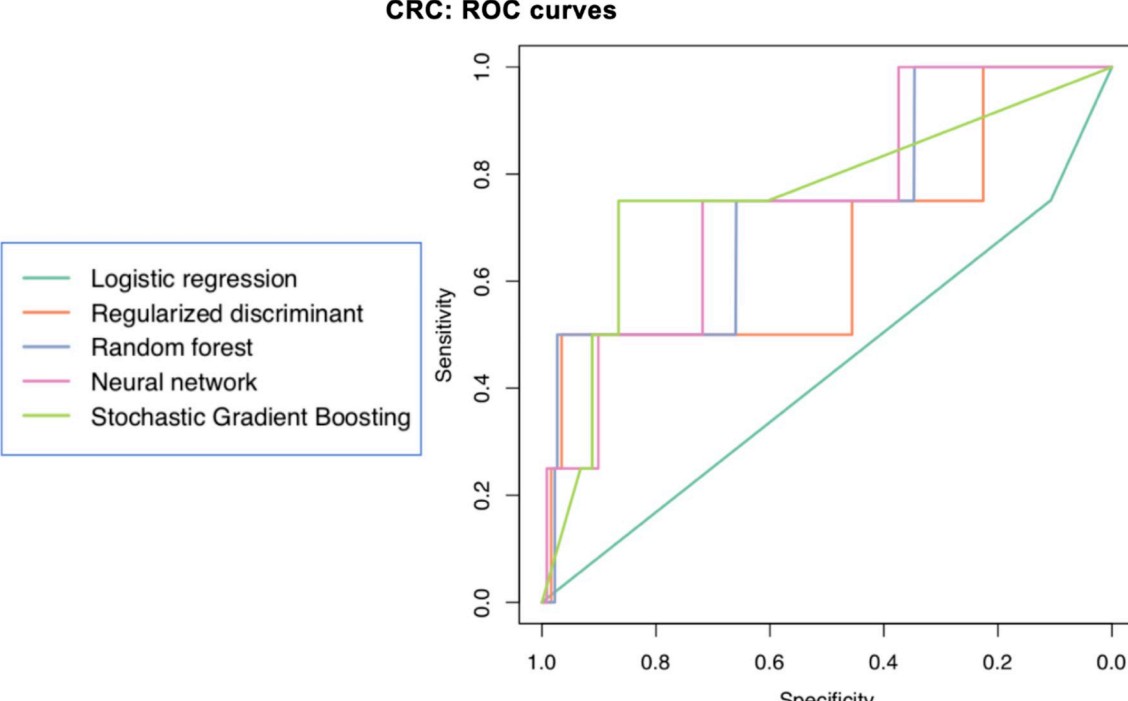

**CRC or high-risk polyps: ROC curves**

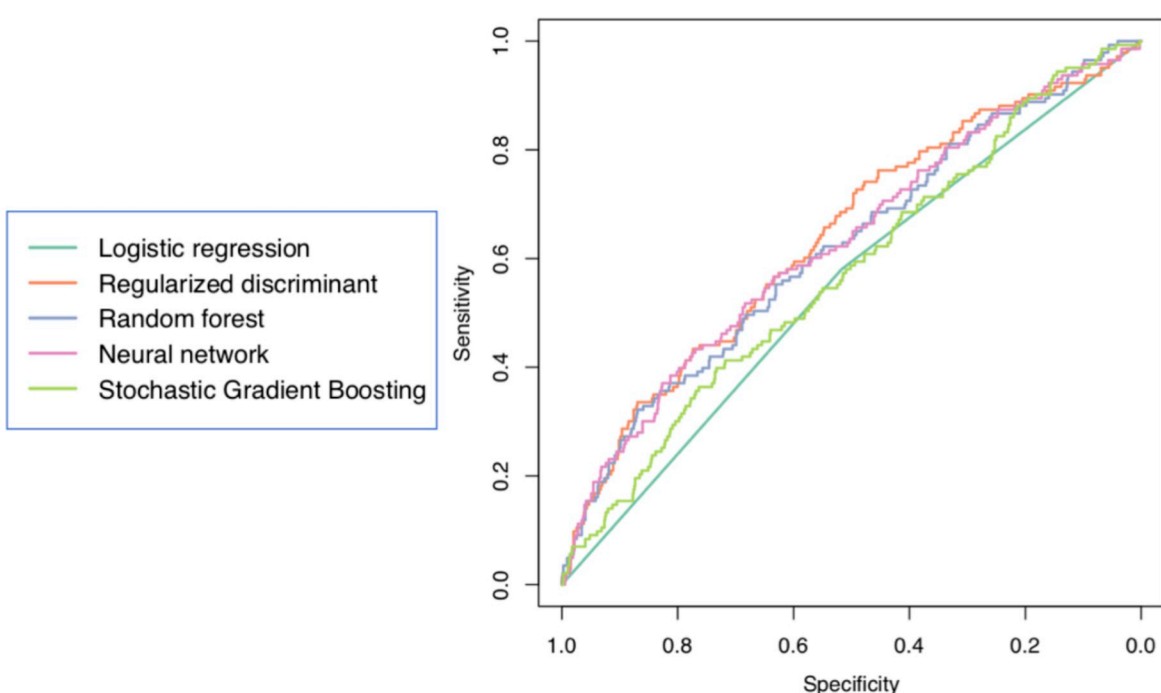

**Fig 2.** Receiver Operator Curves (ROC) of the reference and machine learning models in the test set for colorectal cancer (CRC) and CRC or high-risk polyps (bottom).

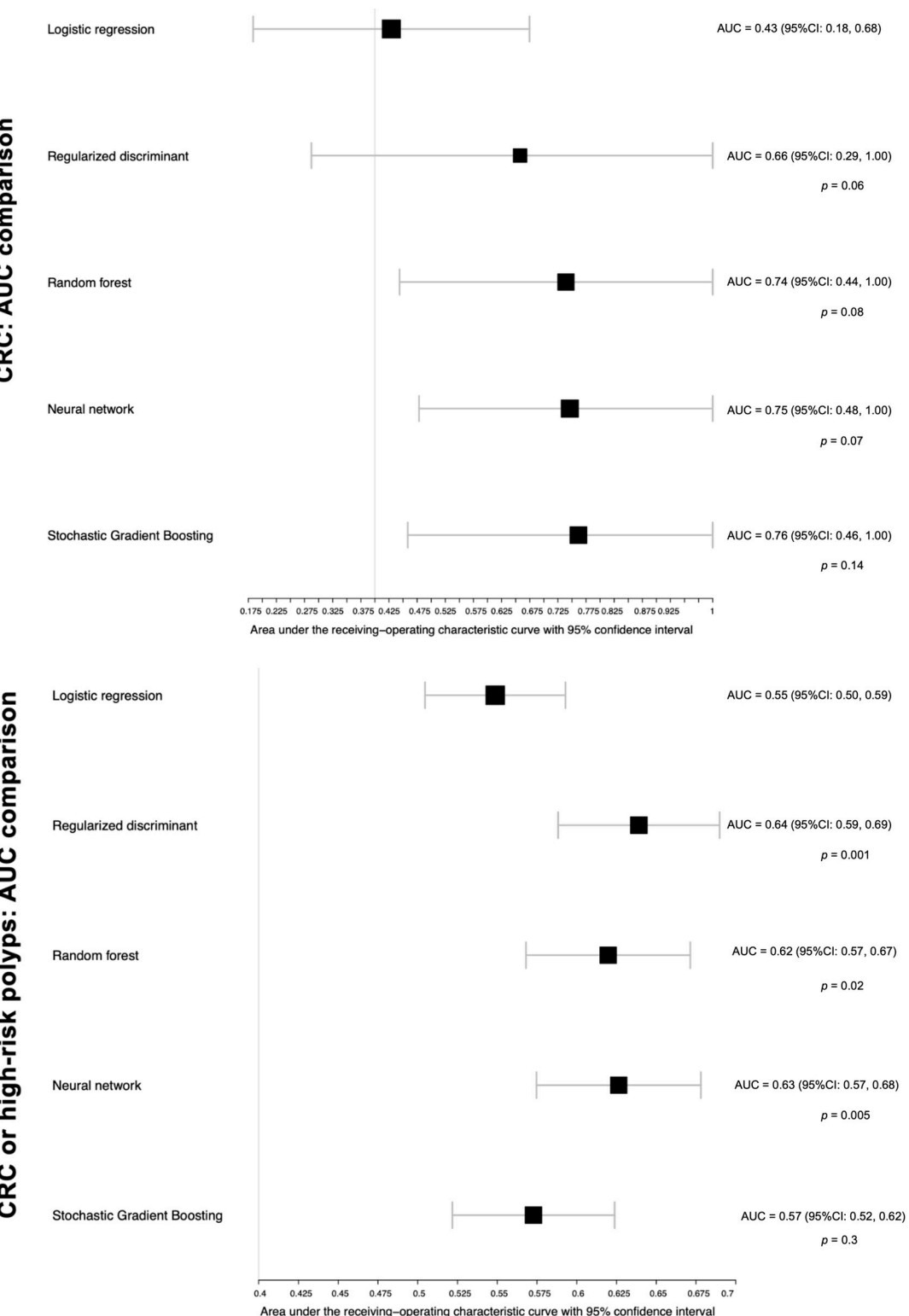

**Fig 3. Area Under the Curve (AUC) of our reference and machine learning models in the test set for colorectal cancer (CRC) and CRC or high-risk polyps.** The *p* value compares the machine learning models to the reference model using the DeLong test.

**Table 2. Performance metrics of different prediction models.**

| Colorectal cancer | | | | | |
|---|---|---|---|---|---|
| Metric/Model | Logistic regression | Regularized discriminant | Random forest | Neural network | Stochastic gradient boosting |
| Accuracy | 0.11 | 0.96 | 0.66 | 0.71 | 0.86 |
| Accuracy (lower) | 0.09 | 0.95 | 0.62 | 0.68 | 0.84 |
| Accuracy (upper) | 0.13 | 0.97 | 0.69 | 0.74 | 0.88 |
| Balanced accuracy | 0.42 | 0.73 | 0.70 | 0.73 | 0.80 |
| Sensitivity | 0.75 | 0.50 | 0.75 | 0.75 | 0.75 |
| Specificity | 0.10 | 0.96 | 0.65 | 0.71 | 0.86 |
| Positive predictive value | 0.00 | 0.05 | 0 | 0.01 | 0.02 |
| Negative predictive value | 0.99 | 0.99 | 0.99 | 0.99 | 0.99 |
| Colorectal cancer or high-risk polyps | | | | | |
| Accuracy | 0.52 | 0.61 | 0.61 | 0.62 | 0.54 |
| Accuracy (lower) | 0.49 | 0.58 | 0.58 | 0.59 | 0.51 |
| Accuracy (upper) | 0.55 | 0.64 | 0.65 | 0.65 | 0.58 |
| Balanced accuracy | 0.54 | 0.6 | 0.59 | 0.60 | 0.54 |
| Sensitivity | 0.58 | 0.57 | 0.55 | 0.56 | 0.54 |
| Specificity | 0.51 | 0.62 | 0.63 | 0.63 | 0.55 |
| Positive predictive value | 0.17 | 0.21 | 0.21 | 0.21 | 0.18 |
| Negative predictive value | 0.87 | 0.89 | 0.88 | 0.89 | 0.87 |

learning for creating a predictive model for colorectal neoplasia using single-center data from 3,116 colonoscopy patients aged 35–50. To develop this model, we carefully selected four machine-learning approaches (regularized discriminant analysis, random forest, neural network, and gradient boosting decision tree) and compared them to a logistic regression model. Regularized discriminant analysis minimizes the misclassification probability compared to logistic regression [41]. Random forest and gradient boosting decision trees are more powerful than logistic regression when there are higher-order interactions between predictors [42, 43]. Neural networks have a high tolerance for noise and are able to diagnose networks on their own [44]. In our analyses, the machine learning models achieved better predictive performance for CRC or high-risk polyps using data routinely available in EHRs (e.g., indication, zip code, BMI, and laboratory studies). The machine learning models also achieved higher specificity, positive predictive values, and accuracy for predicting our outcomes (i.e., leading to less over-utilization of testing). To our knowledge, this is the first study that has applied modern machine learning approaches to predict colorectal neoplasia in adults aged 35–50 with or without symptoms.

Multiple professional societies now recommend initiating CRC screening at age 45 as opposed to 50. However, timely adoption of CRC screening in adults younger than 50 remains challenging, due to the overall low population risk of CRC or pre-malignant polyps, costs of screening, and risks of colonoscopy. An alternative screening strategy is to perform FIT annually starting at 45 years of age followed by colonoscopies starting at 50 years of age. In this scenario, the number of colonoscopies would be reduced, but stool samples would have to be mailed yearly. Ultimately, this cumbersome strategy may not be as cost-effective as expanding screening colonoscopies in older more at-risk adults [6]. Furthermore, despite the high specificity of the FIT for CRC (94.9%), its sensitivity is low for CRC (73.8%), advanced adenomas (23.8%), and advanced serrated lesions (5%) [45].

One example is the Colorectal Risk Assessment Tool (CCRAT), which is endorsed by the National Cancer Institute [46]. CCRAT relies on patients to report risk factors, was only

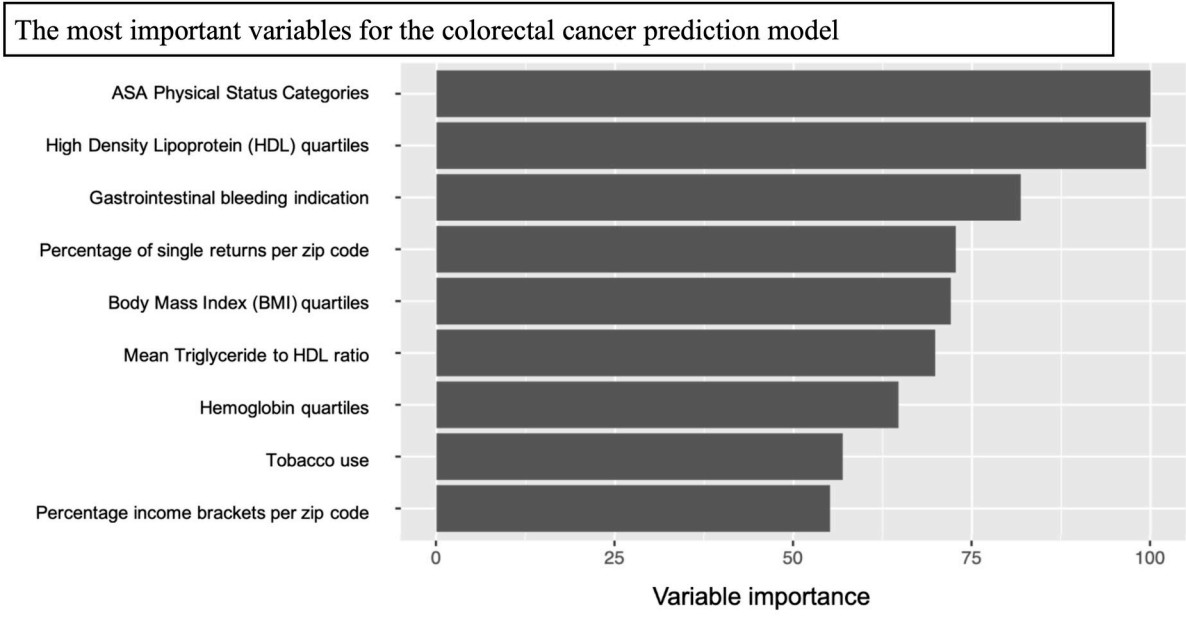

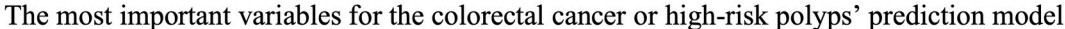

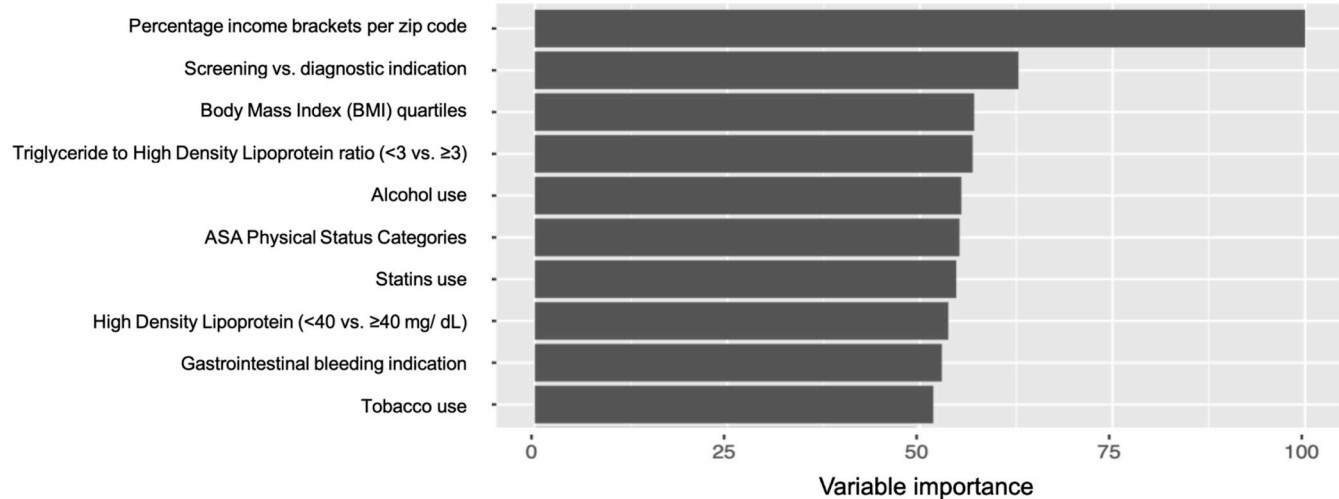

**Fig 4. Comparison of the reference Area Under the Curve (AUC) to machine learning models in the test set for colorectal cancer (CRC) and CRC or high-risk polyps.**

validated in adults over the age of 50, and modestly discriminates CRC (AUC of 0.61) [47]. Although newer tools can stratify the risk of high-risk polyps or CRC, they too have suboptimal discriminatory performance [27]. For instance, a recent head-to-head comparison of 17 risk models yielded an AUC of 0.58 to 0.65 for advanced adenomas or CRC [13]. More recent models had modest prediction ability even when using an extensive list of predictors, which can be cumbersome for patients and providers [27, 48]. Furthermore, most risk stratification tools were created for asymptomatic adults ≥50 years of age and are not specifically tailored for adults <50 years of age or those with symptoms [13]. Indeed, machine learning approaches are reported to be powerful tools for predictive analytics in healthcare [49, 50] and have demonstrated substantial success in many applications such as biomarker identification [51] and

outcome prediction [52, 53]. Therefore, the present study builds on and extends these reports by demonstrating the superior ability of modern machine learning approaches to predict CRC (AUC of 0.75 for the neural network model) and CRC or high-risk polyps (AUC of 0.64 for the regularized discriminant analysis model) compared to conventional logistic regression using variables routinely available in EHRs.

When it comes to practical applications, and based on the improved positive prediction of CRC or high-risk serrated or adenomatous polyps (21.9% in the neural network model compared to conventional regression) observed in our study, we suggest that combination of machine learning-based risk assessment and FIT could offer a cost-effective early screening strategy for adults under the age of 50 and would help to reduce the burden of colonoscopy referrals on the healthcare system. Therefore, future studies should combine machine learning models with non-invasive methods (e.g., FIT and screening for symptoms) to improve the effectiveness of CRC detection in adults under the age of 50. Due to the recent recommendation to lower the age of CRC screening to 45, this approach would be critical to conserve colonoscopy resources by stratifying adults into risk categories.

## Strengths and limitations

Our proof-of-concept study has several strengths and limitations. The strengths include the significance and innovation of the model, rigorous methods and reporting, inclusion of average-age-risk patients with or without symptoms, use of predictors collected during routine clinical care, and internal validation using a split sample. For instance, symptoms (e.g., gastrointestinal bleeding) were important predictors for CRC and can risk-stratify adults in the primary care setting based on symptoms and other CRC predictors. There are several possible explanations for the incremental gains in predictive ability achieved by the machine learning models. For example, although we integrated several well known risk factors for CRC, the categorical formats of continuous variables with clinically meaningful cutoffs may contribute to risk prediction. Machine learning accounts for linear and non-linear relationships between variables, which enhances predictive performance without assuming additivity compared to conventional statistical models [54]. Assembling methods that combine several basic models to produce one optimal model, such as the random forest and gradient boosting decision tree models, results in a well-generalized model and reduces the risk of overfitting [55, 56]. Although machine learning improves predictive ability, the predictions remain imperfect. This is likely due to the subjectivity of symptoms, timing of BMI measurement relative to CRC (early-life vs. pre-diagnosis), and lack of broader predictors, such as diet and physical activity. Although these variables are risk factors for CRC, our objective was to harness the limited set of clinical data that are available in EHRs to develop machine learning models. Machine learning approaches are also data driven and therefore depend on accurate data. This becomes problematic when data is missing. For example, patients referred for screening colonoscopy may have symptoms that were not reported at time of referral [57]. Patients may also under-report recreational drugs' use or the use of over-the-counter medications such as aspirin. Second, the imputation of missing data is a potential source of bias. Nevertheless, imputation by machine learning is a rigorous technique, especially when compared to regression [58]. Third, due to the rarity of CRC, our dataset was imbalanced, which may bias predictions towards the dominant class. We applied the "SMOTE" oversampling method to adjust for this bias when developing our models [40]; however, we anticipate the collection of additional patients with CRC in future studies to better address this issue. Forth, the inclusion of non-high-risk polyps may undermine the discrimination of the models. To evaluate the effect of this inclusion, we compared the sensitivity of the machine learning and regression models after exclusion of

non-high-risk polyps (data not shown). We did not observe an improvement in the predictive power of any of the models.

## Conclusions

In this analysis of data routinely collected in EHRs for clinical purposes, we demonstrated that machine learning has a superior ability to predict the risk of colorectal neoplasia in adults aged 35–50 compared to conventional logistic regression. Our machine learning models improved specificity, positive predictive values, and accuracy compared to logistic regression and therefore have the potential to reduce invasive testing. Future research should aim to validate our model in large primary care and referral settings and to expand machine learning models by using a broader set of predictors. Upon successful completion of this work, machine learning models have the potential to stratify adults aged 35–50 years with or without symptoms into CRC risk categories, which will lead to precise and cost-effective prevention and early detection of CRC. Our ultimate vision is for machine learning risk assessment tools to be seamlessly integrated into the health care electronic medical system for real-time monitoring of patient risk. We expect that using EHR-based risk assessment tool will also reduce barriers to adoption of our model and improve uptake and value of screening.

## Supporting information

**S1 Table. Comparison of predictors in a sub-sample of patients aged 46–49 undergoing diagnostic and screening colonoscopy.**
(DOCX)

## Acknowledgments

We thank Michelle Messer (MM) and Robert Pickard (RP) for their involvement in obtaining the data for the study.

## Author Contributions

**Conceptualization:** Hisham Hussan.

**Data curation:** Jing Zhao.

**Formal analysis:** Jing Zhao.

**Funding acquisition:** Darrell Gray.

**Methodology:** Hisham Hussan, Jing Zhao, Fred Tabung, Qin Ma, Steven K. Clinton.

**Writing – original draft:** Hisham Hussan, Jing Zhao, Abraham K. Badu-Tawiah, Peter Stanich, Fred Tabung, Darrell Gray, Qin Ma, Matthew Kalady, Steven K. Clinton.

**Writing – review & editing:** Hisham Hussan, Jing Zhao, Abraham K. Badu-Tawiah, Peter Stanich, Fred Tabung, Darrell Gray, Qin Ma, Matthew Kalady, Steven K. Clinton.

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
