## [Decision Letter · Decision Letter 0]

28 Jan 2022

PONE-D-21-34521Utility of machine learning
in developing a predictive model for early-age-onset colorectal neoplasia using
electronic health recordsPLOS ONE

Dear Dr. Hussan,

Thank you for submitting your manuscript to PLOS ONE. After careful consideration, we
feel that it has merit but does not fully meet PLOS ONE’s publication criteria as it
currently stands. Therefore, we invite you to submit a revised version of the
manuscript that addresses the points raised during the review process.

Please submit your revised manuscript by Mar 14 2022 11:59PM. If you will need more
time than this to complete your revisions, please reply to this message or contact
the journal office at plosone@plos.org. When
you're ready to submit your revision, log on to https://www.editorialmanager.com/pone/ and select the 'Submissions
Needing Revision' folder to locate your manuscript file.

Please include the following items when submitting your revised
manuscript:A rebuttal letter that responds to each point raised by the academic
editor and reviewer(s). You should upload this letter as a separate file
labeled 'Response to Reviewers'.A marked-up copy of your manuscript that highlights changes made to the
original version. You should upload this as a separate file labeled
'Revised Manuscript with Track Changes'.An unmarked version of your revised paper without tracked changes. You
should upload this as a separate file labeled 'Manuscript'.

If you would like to make changes to your financial disclosure, please include your
updated statement in your cover letter. Guidelines for resubmitting your figure
files are available below the reviewer comments at the end of this letter.

We look forward to receiving your revised manuscript.

Kind regards,

Hsu-Heng Yen

Academic Editor

PLOS ONE

Journal Requirements:

2. Please include the ethics statement in the Methods section, including details of
the name of the ethics committee, the approval number, and the fact that consent was
waived

Reviewers' comments:

Reviewer's Responses to Questions

**Comments to the Author**

1. Is the manuscript technically sound, and do the data support the conclusions?

Reviewer #1: Yes

Reviewer #2: Yes

Reviewer #3: Yes

2. Has the statistical analysis been performed
appropriately and rigorously? 

Reviewer #1: Yes

Reviewer #2: Yes

Reviewer #3: Yes

3. Have the authors made all data underlying the
findings in their manuscript fully available?

Reviewer #1: Yes

Reviewer #2: Yes

Reviewer #3: Yes

4. Is the manuscript presented in an intelligible
fashion and written in standard English?

Reviewer #1: Yes

Reviewer #2: Yes

Reviewer #3: Yes

5. Review Comments to the Author

Reviewer #1: 1. The authors discussing machine learning in this paper but there isn't
any clear definition about it. If the authors can provide a clear definition of
machine learning and how these machine learning have been use at the introduction
section that would helpful

2. As the study involved 3,116 adults aged 35-60, I would be interested to know how
these sample were recruited for the study

3. Overall, the paper has demonstrated clear methodology of research

Reviewer #2: Dear Authors,

The manuscripts is well-organized and well written. The topic is interesting and
significant in the current situation. I would suggest few minor revisions to enhance
the quality of your manuscript.

i) You have added some supporting information. However, the graphs and charts are not
clearly visible. Is it possible to upload better quality image?

ii) The references you used are good enough in quantity, however, it would be
beneficial to add few more references which are up to date (preferably from
2020-2022). It will ensure the research gap.

iii) Please add the managerial and practical significance of the research in a
separate paragraph.

Thank you and good luck!

Reviewer #3: I am pleased to have an opportunity to review this scientific paper
about the predictive models for early-age-onset

colorectal neoplasia. I think this paper is scientifically informative and
interesting. I have some comments and suggestions as below.

1.Major

The author should add the work plot in this manuscript and it's necessary to add the
inclusion and exclusion criteria in this plot.

The figure legend and table should put in the end of manuscript.

You should put the P value in your figure.

The history of colorectal neoplasia in family member may be added in this manuscript,
which was an important factor associated with colorectal cancer.

Patients with abnormal laboratory studies (CEA and CA199) should be excluded in this
manuscript which may result a bias of the cohort.

6. PLOS authors have the option to publish the peer
review history of their article (what does this mean?). If published, this will
include your full peer review and any attached files.

If you choose “no”, your identity will remain anonymous but your review may still be
made public.

**Do you want your identity to be public for this peer review?** For
information about this choice, including consent withdrawal, please see our
Privacy Policy.

Reviewer #1: No

Reviewer #2: **Yes: **Syed Far Abid Hossain

Reviewer #3: No

---

## [Author Response · Author response to Decision Letter 0]

4 Feb 2022

REVIEWERS’ COMMENTS:

Comments to the Author

Reviewer #1: 

1. The authors discussing machine learning in this paper but there isn't any clear
definition about it. If the authors can provide a clear definition of machine
learning and how these machine learning have been use at the introduction section
that would helpful

Answer: Thank you for this important point. We included a definition of machine
learning with a reference in introduction. A description of the utilized machine
learning algorithms is also detailed in the methods and in our discussion. 

2. As the study involved 3,116 adults aged 35-60, I would be interested to know how
these sample were recruited for the study

Answer: This is a very good comment. As described, this is a retrospective cohort
available at the Ohio State University. We clarified further that our cohort and
their colonoscopy data were obtained from the Ohio State University colonoscopy
quality reporting GI Quality Improvement Consortium, Ltd (GIQuic) database. GIQuic
is a collaborative, nonprofit, scientific organization between the American College
of Gastroenterology and the American Society for Gastrointestinal Endoscopy1. We
linked to electronic health records data to our database to form a retrospective
sample of 3,116 adults aged 35-50 with EHR data and colonoscopy-based colorectal
neoplasia outcomes. We clarified this point in our methods and Fig 1. 

3. Overall, the paper has demonstrated clear methodology of research

Answer: Thank you for the kind comment. 

Reviewer #2: 

Dear Authors,

The manuscripts is well-organized and well written. The topic is interesting and
significant in the current situation. I would suggest few minor revisions to enhance
the quality of your manuscript.

1. You have added some supporting information. However, the graphs and charts are not
clearly visible. Is it possible to upload better quality image?

Answer: We apologize the quality. It appears the images lost quality in the
submission process and we have since rectified this mistake.

2. The references you used are good enough in quantity, however, it would be
beneficial to add few more references which are up to date (preferably from
2020-2022). It will ensure the research gap.

Answer: Thank you for pointing that out. We added more up-to-date references. We also
revised and enhanced the quality of the paper. 

3. Please add the managerial and practical significance of the research in a separate
paragraph.

Answer: This is a very reasonable comment. We included practical significance of this
work in one paragraph. 

Thank you and good luck!

Reviewer #3: 

I am pleased to have an opportunity to review this scientific paper about the
predictive models for early-age-onset

colorectal neoplasia. I think this paper is scientifically informative and
interesting. I have some comments and suggestions as below.

1. Major, the author should add the work plot in this manuscript and it's necessary
to add the inclusion and exclusion criteria in this plot.

Answer: Thank you for helping improve the presentation of our paper. We added a plot
describing our inclusion and exclusions (Fig 1).

2. The figure legend and table should put in the end of manuscript.

Answer: Thank you. We are following PLOS ONE's style requirements when it comes to
figure legends and tables. Specifically, per the instructions:

“Each figure caption should appear directly after the paragraph in which they are
first cited.”

“Tables should be included directly after the paragraph in which they are first
cited.” Link to instructions: https://journals.plos.org/plosone/s/file?id=wjVg/PLOSOne_formatting_sample_main_body.pdf

3. You should put the P value in your figure.

Answer: Many thanks. To clarify, the p values compare the machine learning methods to
conventional regressions and are already included in Fig 3. We clarified the meaning
of our p values in Fig 3 in the manuscript. Fig 2 and 4 do not have p values to
insert. 

4. The history of colorectal neoplasia in family member may be added in this
manuscript, which was an important factor associated with colorectal cancer.

Answer: As early-onset CRC primarily occurs in adults with no strong familial
predisposition or pre-existing colitis2, we included only average-risk adults
without family history of CRC. History of colorectal neoplasia was indeed an
included factor in our prediction module, but however it was not high enough
compared to environmental risk factors such as social factors, obesity and insulin
resistance. This is likely since environmental risk factors are likely more
contributing than genetic factors for the increased risk in average-risk adults with
early-onset neoplasia.

5. Patients with abnormal laboratory studies (CEA and CA199) should be excluded in
this manuscript which may result a bias of the cohort.

Answer: We completely agree. None of the patients in the cohort were referred due to
an abnormal CEA or CA199. We clarified that in our exclusions.

REFERENCES:

1. GIQuIC. www.giquic.org. Accessed January 27, 2022.

2. Dozois EJ, Boardman LA, Suwanthanma W, et al. Young-onset colorectal cancer in
patients with no known genetic predisposition: can we increase early recognition and
improve outcome? Medicine (Baltimore). 2008;87(5):259-263.

to reviewers.docx
---

## [Decision Letter · Decision Letter 1]

28 Feb 2022

Utility of machine learning in developing a predictive model for early-age-onset
colorectal neoplasia using electronic health records

PONE-D-21-34521R1

Dear Dr. Hussan,

We’re pleased to inform you that your manuscript has been judged scientifically
suitable for publication and will be formally accepted for publication once it meets
all outstanding technical requirements.

Kind regards,

Hsu-Heng Yen

Academic Editor

PLOS ONE

 Reviewers' comments:

Reviewer's Responses to Questions

**Comments to the Author**

1. If the authors have adequately addressed your comments raised in a previous round
of review and you feel that this manuscript is now acceptable for publication, you
may indicate that here to bypass the “Comments to the Author” section, enter your
conflict of interest statement in the “Confidential to Editor” section, and submit
your "Accept" recommendation.

Reviewer #2: All comments have been addressed

Reviewer #3: All comments have been addressed

2. Is the manuscript technically sound, and do the data
support the conclusions?

Reviewer #2: Yes

Reviewer #3: Yes

3. Has the statistical analysis been performed
appropriately and rigorously? 

Reviewer #2: Yes

Reviewer #3: Yes

4. Have the authors made all data underlying the
findings in their manuscript fully available?

Reviewer #2: Yes

Reviewer #3: Yes

5. Is the manuscript presented in an intelligible
fashion and written in standard English?

Reviewer #2: Yes

Reviewer #3: Yes

6. Review Comments to the Author

Reviewer #2: The revised version is good. Thank you for addressing the comments
properly. The manuscript is well organized now.

Reviewer #3: (No Response)

7. PLOS authors have the option to publish the peer
review history of their article (what does this mean?). If published, this will
include your full peer review and any attached files.

If you choose “no”, your identity will remain anonymous but your review may still be
made public.

**Do you want your identity to be public for this peer review?** For
information about this choice, including consent withdrawal, please see our
Privacy Policy.

Reviewer #2: **Yes: **Syed Far Abid Hossain, Coordinator, MBA Program,
IUBAT

Reviewer #3: No